# HIV/AIDS Mathematical Model of Triangle Transmission

**DOI:** 10.3390/v14122749

**Published:** 2022-12-09

**Authors:** Cristian Camilo Espitia Morillo, João Frederico da Costa Azevedo Meyer

**Affiliations:** 1Universidad de los Llanos, Villavicencio 500003, Colombia; 2Universidade Estadual de Campinas, Campinas 13056-405, Brazil

**Keywords:** HIV/AIDS mathematical model, basic reproduction number, stationary points, local and global stability analysis

## Abstract

In this paper, a mathematical analysis of the HIV/AIDS deterministic model studied in the paper called Mathematical Model of HIV/AIDS Considering Sexual Preferences Under Antiretroviral Therapy, a case study in the previous works preformed by Espitia is performed. The objective is to gain insight into the qualitative dynamics of the model determining the conditions for the persistence or effective control of the disease in the community through the study of basic properties such as positiveness and boundedness; the calculus of the basic reproduction number; stationary points such as disease-free equilibrium (DFE), boundary equilibrium (BE) and endemic equilibrium (EE); and the local stability (LAS) of disease-free equilibrium. The findings allow us to conclude that the best way to reduce contagion and consequently reach a DFE is thought to be the reduction in the rate of homosexual partners, as they are the most affected population by the virus and are therefore the most likely to become infected and spread it. Increasing the departure rate of infected individuals leads to a decrease in untreated infected heterosexual men and untreated infected women.

## 1. Introduction

Epidemiological evidence shows that HIV is transmitted only through the exchange of body fluids such as blood, semen, vaginal or anal secretions, and breast milk. As a result, the highly common means of transmission are: unprotected sex, from mother to child during pregnancy, childbirth or breast feeding, injecting drugs with a needle that has come into contact with infected blood, and infected blood donation or organ transplant [1]. There are many myths and misconceptions about how a person can get HIV. It is not transmitted through body fluids such as sweat, tears, or saliva, touching someone who has HIV, mosquito bites, or other transmission methods.

The sexual transmission of HIV is usually considered to be carried out by heterosexual or homosexual men through anal intercourse. Transmission between two women is almost null; however, this form is possible by sharing toys such as sexual vibrators [2,3]. Female homosexual contact has not been demonstrated to pose appreciable HIV transmission risk, and such transmission appears to be rare [4,5]. According to communication with the HIV/AIDS infectious disease specialist Dr. Alexandre Naime Barbosa, the sexual transmission between men can occur through three mechanisms: exclusive homosexual transmission, exclusive heterosexual transmission, or bisexual transmission, while in women, the transmission is almost always heterosexual. The Center for Disease Control and Prevention estimates that HIV rates in men who have sex with men (MSM) are higher than the rates in heterosexual contacts. In part, these differences reflect the fact that an individual MSM can engage in both insertive and receptive sexual roles (versatility), while exclusively heterosexual men and women each engage in only one of these roles [6,7].

When discussing transmission, the term “Discordant Couples” will be used to represent a couple in which one partner has a sexually transmitted disease while the other partner does not. If two participants are infected, the transmission could imply co-infection, which is not the objective in this investigation. However, if the two participants are susceptible, then there is no contagion.

The risk of acquiring HIV is 22 times higher among men who have sex with men (MSM), 22 times higher among individuals who are injectable drug users (IDU) and share needles, 21 times higher for sex workers, and 12 times higher for transgender people compared to the risk of transmission in heterosexual contact [8,9]. One form of measuring how transmissible a disease is the “Basic Reproduction Number”, which describes secondary infections from a first infection; this number depends on the contagion’s form. For example, for HIV/AIDS transmission, the basic reproduction number is 4 in the homosexual population in the United Kingdom, whereas the basic number is 11 for female prostitutes in Kenya [10]. As a result of the variation in these statistics, we consider homosexual transmission to be greater than heterosexual transmission.

In the triangle transmission model, it is assumed that the only way to transmit the HIV virus is through sexual intercourse, and it is commonly considered that the contagion form takes into account heterosexuals and homosexuals in the dynamic of infection. However, can the population be split into heterosexuals and homosexuals and thus the group of bisexuals be ignored? Moreover, what is the contribution of these group in the transmission of HIV? To try to answer these questions, we propose a different mathematical model considering HIV-infected bisexuals under ART. Several articles have also focused on the whole population of constant size when considering force of infection, although some studies such as [11,12] have stressed the importance of variable population size in epidemic dynamics. All these assumptions, such as sexual preference and variable population in force of infection, are considered in our model.

With regard to sexual contact between homosexual men, heterosexual men and women [13,14] say: “There exist individuals that change their sexual behavior depending on the situation or at different stages in their life. A possibly common and transient example of situational sexuality is the person who self-identifies as heterosexual, but will sexually interact with a member of the same sex when lacking other opportunities. Less transient but also possibly common, a person who self-identifies as gay or lesbian (either at the time, or later) may sexually interact with a member of the opposite sex if a same-sex relationship seems unfeasible”. Thus, in our model, we consider bisexual contact.

## 2. Materials and Methods

The epidemiological model under consideration was studied in [15]. The model contains three population groups: the first being men with homosexual preference in men, the second being men with heterosexual preference, and the third one for women who may be homosexual or heterosexual but engage in sexual relations with homosexual or heterosexual men. We supposed that, eventually, the homosexual men had sexual contact with women and that the heterosexual men had sexual relation with homosexual men. Consequently, we consider bisexual behavior among these groups because the transmission from homosexuals to heterosexual men or women goes through the bisexuals. Female homosexual transmission is not considered in the dynamic of infection. For more information, see references [2,3].

The total population N(t) is divided into eight classes; $S_{h}\left(t\right)$ represents susceptible homosexual men, $I_{h}\left(t\right)$ untreated infected homosexual men, $S_{w}\left(t\right)$ susceptible women, $I_{w}\left(t\right)$ untreated infected women, $S_{m}\left(t\right)$ susceptible heterosexual men, $I_{m}\left(t\right)$ untreated infected heterosexual men, T(t) treated individuals on ART, and A(t) individuals living with AIDS.

Figure 1 represents the transmission dynamics between the three studied sexual preferences. Each vertex of the triangle represents one population, and the sides of the triangle denote the different forms of transmission between the populations involved. To begin, the exclusive transmission among homosexual men is illustrated by the upper circular dotted arrow labeled as $\lambda_{h}$. Then, the transmission between homosexual and heterosexual men and the transmission between homosexual men and women are represented by dashed lines identified as $\lambda_{hm}$ and $\lambda_{hw}$, respectively. Finally, heterosexual transmission between men and women is a continuous line represented by $\lambda_{m,w}$. The direction of the arrows represents the sense of the analyzed contagion; nonetheless, contagions can biologically occur in all directions. Consequently, the two following hypotheses are assumed: the only form of contagion among homosexuals is among themselves, and heterosexual people become infected due to the contact with homosexual men or heterosexual partners of the opposite sex. Thus, dashed lines have only one direction, while the continuous line between heterosexual men and women has two directions. The following assumed hypotheses in the model were evaluated by HIV/AIDS specialist Dr. Alexandre Naime Barbosa from Stadual University of Sao Paulo, UNESP, Botocatu, Brazil.

Assumed Hypotheses in the Model.

**H1** Constant recruitment in all susceptible classes is assumed.**H2** Sexual transmission in discordant couples is considered.**H3** Homosexual individuals become infected among themselves. HIV transmission in the susceptible female population happens through sexual relations with infected heterosexual men or with infected homosexual men. Susceptible heterosexual men can become infected by infected women or infected homosexual men.**H4** There is no gender differentiation in either sexual preference in treated individuals or individuals living with AIDS.**H5** Individuals living with AIDS could be treated or untreated, noting that an individual that developed AIDS during a hospital treatment will be diagnosed and enrolled in ART.**H6** It is considered both natural mortality in all classes and induced mortality in individuals living with AIDS.

Parameters in the Model

The constant recruitment in all susceptible classes is denoted by Ψ. The male proportion is labeled by θ, 0≤θ≤1. The heterosexual proportion is represented by γ, 0≤γ≤1. The proportion of initially treated individuals is *p*, 0≤p≤1; consequently, (1−p) denotes the proportion of untreated individuals. Natural mortality rate is symbolized by μ. Induced mortality rate in individuals living with AIDS is *d*. AIDS development rate in treated individuals is δ. Departure rate of infected individuals is α. Subscripts s,h,hw,hm mean sexual contact between heterosexual men and women, among homosexual men, between homosexual men and women, and, finally, between homosexual men and heterosexual men, respectively; thus, $\beta_{s,h,hw,hm}$ represents the probability of transmission and $c_{s,h,hw,hm}$ mean rate of sexual partners in the aforementioned contacts. $B_{h}=c_{h}\beta_{h},B_{s}=c_{s}\beta_{s},B_{hm}=c_{hm}\beta_{hm},B_{hw}=c_{hw}\beta_{hw}$ rates will be considered for parameter simplification. All parameters are non-negatives and are listed in Table 1.

Initially treated individuals and individuals living with AIDS receiving ART are disregarded from the transmission because their viral load is negligible. In Figure 1, we assume that susceptible homosexual men only become infected by infected homosexual men, and susceptible women (or men) become infected by infected men (or women) or infected homosexual men. This means that susceptible homosexual men select their partner randomly from the infected homosexual population, while women or men select their partners randomly from the infected heterosexual or infected homosexual population [15].

The force of infection or disease incidence function measures the susceptible person’s risk of becoming infected. In some epidemic models, this function is assumed to be bilinear in both the infected individuals and the susceptible individuals. In addition, a bilinear force of infection or mass action law incidence may not yield appropriate results for several reasons. In particular, this force of infection does not permit one to consider the difference among infected individuals. Thus, we decided that since this function represents the contact between an infected person and a susceptible one, the denominator would have to only be formed by susceptible individuals and those who can transmit the disease. We excluded both treated individuals and people living with AIDS under ART since their viral charge is negligible; in addition, people living with AIDS are too sick, and their sexual life can be considered as almost null. Therefore, the following infection forces by sexual contact are:$$\begin{array}{cccc} \lambda_{h} & =B_{h}\frac{I_{h}}{S_{h}+I_{h}} & \; & \mathrm{Exclusive}\;\mathrm{Homosexual}\;\mathrm{Contact},\\ \lambda_{hw} & =B_{hw}\frac{I_{h}}{S_{m}+S_{h}+I_{m}+I_{h}} & \; & \mathrm{Contact}\;\mathrm{between}\;\mathrm{Homosexual}\;\mathrm{Men}\;\mathrm{and}\;\mathrm{Women},\\ \lambda_{hm} & =B_{hm}\frac{I_{h}}{S_{m}+S_{h}+I_{m}+I_{h}} & \; & \mathrm{Contact}\;\mathrm{between}\;\mathrm{Homosexual}\;\mathrm{Men}\;\mathrm{and}\;\mathrm{Heterosexual}\;\mathrm{Men},\\ \lambda_{m,w} & =B_{s}\frac{I_{m,w}}{S_{m,w}+I_{m,w}} & \; & \mathrm{Heterosexual}\;\mathrm{Contact}. \end{array}$$

It is important to note that in exclusive homosexual contact, the fraction denotes untreated infected homosexual men among susceptible and untreated infected homosexual men. However, in the contact between homosexual men and women, the fraction denotes untreated infected homosexual men among susceptible heterosexual men and untreated homosexual men because this contact is considered bisexual behavior. The same reasoning should be applied to the contact between homosexual men and heterosexual men. For heterosexual contact, the fraction denotes untreated infected heterosexual men (women) among susceptible heterosexual men (women) and untreated infected heterosexual men (women). The compartmental model is presented in Figure 2. The dynamic is governed by the system of nonlinear ordinary differential Equations (Equation 1)–(Equation 8), where a dot represents differentiation with respect to *t*.
(1)$$\begin{array}{cc} \dot{S_{h}} & =\Psi \theta \left(1-\gamma \right)-B_{h}\frac{I_{h}}{S_{h}+I_{h}}S_{h}-\mu S_{h}, \end{array}$$
(2)$$\begin{array}{cc} \dot{I_{h}} & =B_{h}\frac{I_{h}}{S_{h}+I_{h}}S_{h}-\left(\alpha +\mu \right)I_{h}, \end{array}$$
(3)$$\begin{array}{cc} \dot{S_{w}} & =\Psi \left(1-\theta \right)-B_{s}\frac{I_{m}}{S_{m}+I_{m}}S_{w}-B_{hw}\frac{I_{h}}{S_{m}+S_{h}+I_{m}+I_{h}}S_{w}-\mu S_{w}, \end{array}$$
(4)$$\begin{array}{cc} \dot{I_{w}} & =B_{s}\frac{I_{m}}{S_{m}+I_{m}}S_{w}+B_{hw}\frac{I_{h}}{S_{m}+S_{h}+I_{m}+I_{h}}S_{w}-\left(\alpha +\mu \right)I_{w}, \end{array}$$
(5)$$\begin{array}{cc} \dot{S_{m}} & =\Psi \theta \gamma -B_{s}\frac{I_{w}}{S_{w}+I_{w}}S_{m}-B_{hm}\frac{I_{h}}{S_{m}+S_{h}+I_{m}+I_{h}}S_{m}-\mu S_{m}, \end{array}$$
(6)$$\begin{array}{cc} \dot{I_{m}} & =B_{s}\frac{I_{w}}{S_{w}+I_{w}}S_{m}+B_{hm}\frac{I_{h}}{S_{m}+S_{h}+I_{m}+I_{h}}S_{m}-\left(\alpha +\mu \right)I_{m}, \end{array}$$
(7)$$\begin{array}{cc} \dot{T} & =\alpha p\left(I_{h}+I_{w}+I_{m}\right)-\left(\delta +\mu \right)T, \end{array}$$
(8)$$\begin{array}{cc} \dot{A} & =\alpha \left(1-p\right)\left(I_{h}+I_{w}+I_{m}\right)+\delta T-\left(d+\mu \right)A. \end{array}$$

With initial conditions
$$\begin{array}{cc} S_{h}\left(0\right) & >0,\;I_{h}\left(0\right)\geq 0\;S_{w}\left(0\right)>0,\;I_{w}\left(0\right)\geq 0,\;S_{m}\left(0\right)>0,\\ I_{m}\left(0\right) & \geq 0,\;T(0)\geq 0,\;A(0)\geq 0. \end{array}$$

Explanation of Equations

Susceptible individuals such as homosexual men, women, and heterosexual men $S_{h}\left(t\right),S_{w}\left(t\right)$, and $S_{m}\left(t\right)$, grow in number with recruitment Ψθ(1−γ),Ψ(1−θ), and Ψθγ, respectively, where Ψ is a constant recruitment, θ is the male proportion, and γ is the heterosexual proportion; these susceptible populations decrease due to contagion with the virus in contact rates $\lambda_{h},\lambda_{m}$, and $\lambda_{w}$, respectively. Women and heterosexual men additionally acquire the virus with rate $\lambda_{hw}$ and $\lambda_{hm}$. Finally, they can die from natural causes with rate μ.

The number of infected individuals such as homosexual men, women, and heterosexual men, $I_{h}\left(t\right),$$I_{w}\left(t\right)$, and $I_{m}\left(t\right)$, grows with the rates of infection $\lambda_{h},\lambda_{m}$, and $\lambda_{w}$. However, women and heterosexual men grow with rates $\lambda_{hw}$ and $\lambda_{hm}$, respectively. This infected population reduces because its individuals become treated or as a result of people living with AIDS in rates α and α(1−p), respectively. Finally, they die from natural causes with rate μ.

The number of treated individuals, T(t), grows because infected ones enroll in ART, develop AIDS with a rate δ, or die from natural causes with rate μ.

The number of individuals living with AIDS, A(t), grows due of the entrance of infected people with or without treatment whom develop AIDS; they die from natural causes with rate μ and from induced disease death with rate *d*.

The correspondent mathematical analysis of this ordinary differential equations system is developed as follows.

### 2.1. Positiveness and Boundedness

**Theorem** **1.**
*Let the initial conditions be $S_{h}\left(0\right)>0,I_{h}\left(0\right)\geq 0,S_{w}\left(0\right)\geq 0,I_{w}\left(0\right)\geq 0,S_{m}\left(0\right)\geq 0,I_{m}\left(0\right)\geq 0,\;T\left(0\right)\geq 0,\;A\left(0\right)\geq 0$. Then, the solutions $S_{h}\left(t\right),\;I_{h}\left(t\right),\;S_{w}\left(t\right),$
$I_{w}\left(t\right),\;S_{m}\left(t\right),\;I_{m}\left(t\right),\;T\left(t\right),\;A(t)$ of the system *(Equation 1)* to *(Equation 8)* will be positive for all time t>0.*


**Proof.** Let $t_{1}=\sup\left\{t>0:S_{h}\left(t\right)>0,I_{h}\left(t\right)>0,S_{w}\left(t\right)>0,I_{w}\left(t\right)>0,S_{m}\left(t\right)>0,I_{m}\left(t\right)>0,T\left(t\right)>0,A\left(t\right)>0\right\}$. From the first Equation (Equation 1), we have
$$\frac{dS_{h}}{dt}\left(t\right)=\Psi \theta \left(1-\gamma \right)-B_{h}\frac{I_{h}\left(t\right)}{S_{h}\left(t\right)+I_{h}\left(t\right)}S_{h}\left(t\right)-\mu S_{h}\left(t\right)=\Psi \theta \left(1-\gamma \right)-\left(\lambda_{h}\left(t\right)+\mu \right)S_{h}\left(t\right),$$
which can be re-written as:
$$\begin{array}{cc} \frac{d}{dt}(S_{h}\left(t\right)\exp[\mu t+\int_{0}^{t}\lambda_{h}\left(\tau \right)d\tau ]) & =\Psi \theta \left(1-\gamma \right)\exp[\mu t+\int_{0}^{t}\lambda_{h}\left(\tau \right)d\tau ]\\ S_{h}\left(t_{1}\right)\exp[\mu t_{1}+\int_{0}^{t_{1}}\lambda_{h}\left(\tau \right)d\tau ] & -S_{h}\left(0\right)=\Psi \theta \left(1-\gamma \right)\int_{0}^{t_{1}}\exp[\mu y+\int_{0}^{y}\lambda_{h}\left(\tau \right)d\tau ]dy \end{array}$$
$$\begin{array}{cc} S_{h}\left(t_{1}\right) & =S_{h}\left(0\right)\exp[-\mu t_{1}-\int_{0}^{t_{1}}\lambda_{h}\left(\tau \right)d\tau ]\\ \; & +\exp[-\mu t_{1}-\int_{0}^{t_{1}}\lambda_{h}\left(\tau \right)d\tau ]\Psi \theta \left(1-\gamma \right)\int_{0}^{t_{1}}\exp[\mu y+\int_{0}^{y}\lambda_{h}\left(\tau \right)d\tau ]dy\geq 0. \end{array}$$Similarly, it can be shown that $I_{h}\left(t\right),S_{w}\left(t\right),I_{w}\left(t\right),S_{m}\left(t\right),I_{m}\left(t\right),T\left(t\right),A\left(t\right)$ are non-negatives for all time t>0. In this way, all solutions of the system remain positive for all non-negative initial conditions. □

**Theorem** **2.**
*All the solutions of the system *(Equation 1)* to *(Equation 8)* are uniformly bounded. It means any trajectory that starts in $R_{8}^{+}$ remains in $R_{8}^{+}$ for all time t≥0.*


**Proof.** Adding all eight equations from (Equation 1) to (Equation 8) gives:
$$\begin{array}{cc} \frac{dN}{dt} & =\Psi -\mu N-dA\\ \; & \leq \Psi -\mu N. \end{array}$$Solving the differential in-equation, we have:
(9)$$N\left(t\right)\leq (N\left(0\right)-\frac{\Psi }{\mu })\exp\left(-\mu t\right)+\frac{\Psi }{\mu }.$$Therefore, all solutions of the system will enter into the region:
(10)$$\Omega_{III}=\left\{(S_{h}\left(t\right),I_{h}\left(t\right),S_{w}\left(t\right),I_{w}\left(t\right),S_{m}\left(t\right),I_{m}\left(t\right),T\left(t\right),A\left(t\right))\in R_{8}^{+}:N\left(t\right)\leq \frac{\Psi }{\mu }\right\}.$$In Equation (Equation 9), if $N\left(0\right)\leq \frac{\Psi }{\mu }$, then $N\left(t\right)\leq \frac{\Psi }{\mu }$; if $N\left(0\right)\geq \frac{\Psi }{\mu }$ then either the solution enters in $\Omega_{III}$ in finite time or N(t) approaches $\frac{\Psi }{\mu }$ asymptotically. Therefore, $\Omega_{III}$ attracts all solutions in $R_{+}^{8}$. □

The previous theorems allow us to conclude that the region $\Omega_{III}$ is a positively invariant set.

### 2.2. Basic Reproduction Number

The basic reproduction number, $R_{0}$, determines the ability of the virus to develop and persist in the population. It is the average number of individuals that a single infected individual can infect during their life time when introduced into a wholly susceptible population. If $R_{0}<1$, then, on average, a few infected individuals brought into a fully susceptible population will not be able to replace themselves and the disease will not spread. If $R_{0}>1$, then the number of infected individuals will increase with each generation and the disease will spread.

In this research, we use the next generation matrix method as presented in [16]. This method is as follows:

Let $x={\left(x_{1},x_{2},\ldots ,x_{n}\right)}^{T}$ be the number of individuals in each compartment, where the first m<n compartments contain infected individuals. Consider these equations written in the form:(11)$$\dot{x_{i}}=f_{i}\left(x\right)=F_{i}\left(x\right)-V_{i}\left(x\right),\;\mathrm{for}\;i=1,\ldots ,m.$$

In this splitting, $F_{i}\left(x\right)$ is the rate of appearance of new infections in compartment *i* and $V_{i}\left(x\right)=V_{i}^{-}\left(x\right)-V_{i}^{+}\left(x\right)$, where $V_{i}^{+}\left(x\right)$ is the rate of transfer of individuals into compartment *i* by all others, and $V_{i}^{-}\left(x\right)$ is the rate of transfer of individuals out of the *i* compartment.

Note that $F_{i}\left(x\right)$ includes only infections that are newly arising, but does not include terms which describe the transfer of infectious individuals from one compartment to another. Let $X_{s}=\left\{x\geq 0\;|\;x_{i}=0,\;i=1,\ldots ,m\right\}$ be the DFE. Assume that $F_{i}$ and $V_{i}$ satisfy the following axioms outlined by [16]:$\left(A_{1}\right)$If x≥0, then $F_{i},\;V_{i}^{+},\;V_{i}^{-}\geq 0$ for i=1,…,m.$\left(A_{2}\right)$If $x_{i}=0$, then $V_{i}^{-}=0$. In particular, if $x\in X_{s}$, then $V_{i}^{-}=0$ for i=1,…,m.$\left(A_{3}\right)$$F_{i}=0$ if i>m;$\left(A_{4}\right)$If $x\in X_{s}$, then $F_{i}\left(x\right)=0$ and $V_{i}^{+}=0$ for i=1,…,m.$\left(A_{5}\right)$All eigenvalues of $Df(x_{0})$ have negative real parts, where $Df(x_{0})$ is the Jacobian matrix evaluated at the disease free equilibrium $x_{0}$.

**Theorem** **3**(Exposed in [16])**.** *If $x_{0}$ is the disease free equilibrium (DFE) and $f_{i}\left(x\right)$ satisfies $\left(A_{1}\right)-\left(A_{5}\right)$, then the derivatives $DF(x_{0})$ and $DV(x_{0})$ are partitioned as:*
$$DF\left(x_{0}\right)=\left(\begin{array}{cc} F & 0\\ 0 & 0 \end{array}\right),\;DV\left(x_{0}\right)=\left(\begin{array}{cc} V & 0\\ J_{3} & J_{4} \end{array}\right).$$
*where F and V are the m×m matrices defined by:*
$$F=\left[\frac{\partial F_{i}}{\partial x_{j}}\left(x_{0}\right)\right],\;V=\left[\frac{\partial V_{i}}{\partial x_{j}}\left(x_{0}\right)\right]\;\mathrm{with}\;1\leq i,j\leq m.$$
*Furthermore, F is non-negative, V is a non-singular M−matrix, and all eigenvalues of $J_{4}$ have a positive real part.*


According to [17], $FV^{-1}$ is called the next generation matrix for model (Equation 11), and the spectral radius (dominant eigenvalue) is the basic reproduction number:(12)$$R_{0}=\rho \left(FV^{-1}\right).$$

**Theorem** **4**(Exposed in [16])**.** *Consider the disease transmission model given by *(Equation 11)* with f(x) satisfying conditions $\left(A_{1}\right)$ to $\left(A_{5}\right)$. If $x_{0}$ is a DFE of the model, then $x_{0}$ is locally asymptotically stable if $R_{0}<1$, but unstable if $R_{0}>1$, where $R_{0}$ is defined by Equation* (Equation 12)*.*

The basic reproduction number is defined as the spectral radius of the matrix $FV^{-1}$ and denoted by:(13)$$R_{0}=\max\;\left\{\frac{B_{s}}{\alpha +\mu },\frac{B_{h}}{\alpha +\mu }\right\}=\max\left\{R_{0}^{het},R_{0}^{hom}\right\}.$$

Details are presented in Appendix A.

### 2.3. Stationary Points

To calculate stationary points, we solve the associated homogeneous system (Equation 14)–(Equation 21), state variables with a star (*) superscript will be assumed to be an equilibrium value: (14)$$\begin{array}{cc} 0 & =\Lambda_{h}-B_{h}\frac{I_{h}^{*}}{S_{h}^{*}+I_{h}^{*}}S_{h}^{*}-\mu S_{h}^{*}, \end{array}$$(15)$$\begin{array}{cc} 0 & =B_{h}\frac{I_{h}^{*}}{S_{h}^{*}+I_{h}^{*}}S_{h}^{*}-\left(\alpha +\mu \right)I_{h}^{*}, \end{array}$$(16)$$\begin{array}{cc} 0 & =\Lambda_{w}-B_{s}\frac{I_{m}^{*}}{S_{m}^{*}+I_{m}^{*}}S_{w}^{*}-B_{hw}\frac{I_{h}^{*}}{S_{m}^{*}+S_{h}^{*}+I_{m}^{*}+I_{h}^{*}}S_{w}^{*}-\mu S_{w}^{*}, \end{array}$$(17)$$\begin{array}{cc} 0 & =B_{s}\frac{I_{m}^{*}}{S_{m}^{*}+I_{m}^{*}}S_{w}^{*}+B_{hw}\frac{I_{h}^{*}}{S_{m}^{*}+S_{h}^{*}+I_{m}^{*}+I_{h}^{*}}S_{w}^{*}-\left(\alpha +\mu \right)I_{w}^{*}, \end{array}$$(18)$$\begin{array}{cc} 0 & =\Lambda_{m}-B_{s}\frac{I_{w}^{*}}{S_{w}^{*}+I_{w}^{*}}S_{m}^{*}-B_{hm}\frac{I_{h}^{*}}{S_{m}^{*}+S_{h}^{*}+I_{m}^{*}+I_{h}^{*}}S_{m}^{*}-\mu S_{m}^{*}, \end{array}$$(19)$$\begin{array}{cc} 0 & =B_{s}\frac{I_{w}^{*}}{S_{w}^{*}+I_{w}^{*}}S_{m}^{*}+B_{hm}\frac{I_{h}^{*}}{S_{m}^{*}+S_{h}^{*}+I_{m}^{*}+I_{h}^{*}}S_{m}^{*}-\left(\alpha +\mu \right)I_{m}^{*}, \end{array}$$(20)$$\begin{array}{cc} 0 & =\alpha p\left(I_{h}^{*}+I_{w}^{*}+I_{m}^{*}\right)-\left(\delta +\mu \right)T^{*}, \end{array}$$(21)$$\begin{array}{cc} 0 & =\alpha \left(1-p\right)\left(I_{h}^{*}+I_{w}^{*}+I_{m}^{*}\right)+\delta T^{*}-\left(d+\mu \right)A^{*}. \end{array}$$
where
$$\begin{array}{cc} \Lambda_{h} & =\Psi \theta \left(1-\gamma \right),\;\Lambda_{w}=\Psi \left(1-\theta \right),\;\Lambda_{m}=\Psi \theta \gamma ,\;B_{h}=c_{h}\beta_{h},\;B_{s}=c_{s}\beta_{s},\\ B_{hm} & =c_{hm}\beta_{hm},\;B_{hw}=c_{hw}\beta_{hw}. \end{array}$$

Thus, stationary points are:Disease-Free EquilibriumThis happens when $I_{h}^{*}=I_{w}^{*}=I_{m}^{*}=0$ and represents absence of infection. It is:
(22)$$E_{0}=\left(\frac{\Lambda_{h}}{\mu },0,\frac{\Lambda_{w}}{\mu },0,\frac{\Lambda_{m}}{\mu },0,0,0\right).$$Boundary EquilibriumThis occurs when $I_{h}^{*}=0$, the male homosexual population is null, and $I_{w}^{*},\;I_{m}^{*}$ are non-zero. The subscript ($\overset{¯*}{\;}$) means the boundary equilibrium coordinate, which is:
$$E_{1}=\left(\frac{\Lambda_{h}}{\mu },0,\bar{S_{w}^{*}},\bar{I_{w}^{*}},\bar{S_{m}^{*}},\bar{I_{m}^{*}},\bar{T^{*}},\bar{A^{*}}\right),\;\mathrm{where}$$
(23)$$\begin{array}{cccc} \bar{S_{w}^{*}} & =\frac{\Lambda_{w}}{B_{s}-\alpha }, & \bar{S_{m}^{*}} & =\frac{\Lambda_{m}}{B_{s}-\alpha },\\ \bar{I_{w}^{*}} & =\frac{\Lambda_{w}}{B_{s}-\alpha }\left[R_{0}^{het}-1\right], & \bar{I_{m}^{*}} & =\frac{\Lambda_{m}}{B_{s}-\alpha }\left[R_{0}^{het}-1\right],\\ \bar{T^{*}} & =\frac{\Psi p\alpha \left[1-\theta (1-\gamma )\right]}{\left(B_{s}-\alpha \right)\left(\delta +\mu \right)}\left[R_{0}^{het}-1\right], & \bar{A^{*}} & =\frac{\Psi \alpha \left[1-\theta (1-\gamma )\right]\left[\delta +\mu (1-p)\right]}{\left(B_{s}-\alpha \right)\left(\delta +\mu \right)\left(d+\mu \right)}\left[R_{0}^{het}-1\right]. \end{array}$$Note that the boundary equilibrium only exists when $R_{0}^{het}>1$ (implying $B_{s}>\alpha$).Endemic EquilibriumThis represents persistence of the infection, it is:
(24)$$E_{2}=\left(S_{h}^{*},I_{h}^{*},S_{w}^{*},I_{w}^{*},S_{m}^{*},I_{m}^{*},T^{*},A^{*}\right),\;\mathrm{where}$$
(25)$$\begin{array}{cc} S_{h}^{*}= & \frac{\Lambda_{h}}{B_{h}-\alpha },\;\;\;\;\;I_{h}^{*}=\frac{\Lambda_{h}}{B_{h}-\alpha }\left[R_{0}^{hom}-1\right],\\ S_{w}^{*}= & \frac{\Lambda_{w}-\left(\alpha +\mu \right)I_{w}^{*}}{\mu },\;\;S_{m}^{*}=\frac{\Lambda_{m}-\left(\alpha +\mu \right)I_{m}^{*}}{\mu },\\ T^{*}= & \frac{\alpha p\left[\Lambda_{h}\left(R_{0}^{hom}-1\right)+\left(B_{h}-\alpha \right)\left(I_{w}^{*}+I_{m}^{*}\right)\right]}{\left(B_{h}-\alpha \right)\left(\delta +\mu \right)},\\ A^{*}= & \frac{\alpha \left(\delta +\mu (1-p)\right)\left[\Lambda_{h}\left(R_{0}^{hom}-1\right)+\left(B_{h}-\alpha \right)\left(I_{w}^{*}+I_{m}^{*}\right)\right]}{\left(B_{h}-\alpha \right)\left(\delta +\mu \right)\left(d+\mu \right)}. \end{array}$$
(26)$$I_{w}^{*}=\frac{\Lambda_{w}\left(\alpha I_{m}^{*}-\Lambda_{m}\right)[B_{s}I_{m}^{*}\left(B_{h}-\alpha \right)+B_{hw}\Lambda_{h}\left(R_{0}^{hom}-1\right)]-B_{s}\Lambda_{h}\Lambda_{w}\mu R_{0}^{hom}I_{m}^{*}}{\left(\alpha +\mu \right)\{(\alpha I_{m}^{*}-\Lambda_{m})[\left(B_{h}-\alpha \right)(I_{m}^{*}\left(B_{s}-\alpha \right)+\Lambda_{m})+B_{hw}\Lambda_{h}\left(R_{0}^{hom}-1\right)]-\Lambda_{h}\mu R_{0}^{hom}(I_{m}^{*}\left(B_{s}-\alpha \right)+\Lambda_{m})\}}.$$

$I_{m}^{*}$ is given by the roots of fourth degree polynomial:(27)$$p\left(I_{m}^{*}\right)=a_{4}{\left(I_{m}^{*}\right)}^{4}+a_{3}{\left(I_{m}^{*}\right)}^{3}+a_{2}{\left(I_{m}^{*}\right)}^{2}+a_{1}\left(I_{m}^{*}\right)+a_{0}.$$

Coefficients $a_{0},a_{1},a_{2},a_{3}$, and $a_{4}$ are shown in Appendix B.

EE exists when $R_{0}^{hom}>1$. For infected males and females, the following inequalities (Equation 28) must be satisfied. Otherwise, the populations of susceptible men $S_{m}^{*}$ and women $S_{w}^{*}$ will be negatives:(28)$$0<I_{m}^{*}<\frac{\Lambda_{m}}{\alpha +\mu }\;\mathrm{and}\;0<I_{w}^{*}<\frac{\Lambda_{w}}{\alpha +\mu }.$$

Figure 3 shows the existence of equilibrium points, such as Disease-Free Equilibrium (DFE), Boundary Equilibrium (BE), and Endemic equilibrium (EE) in function of $R_{0}^{het}$ and $R_{0}^{hom}$. The figure shows two important aspects. First, the DFE is the only stationary point that exists when $R_{0}^{hom}$ or $R_{0}^{het}$ are less than one; it gives an idea of how stability can be. Second, for existence of EE, the $R_{0}^{hom}$ is more important that $R_{0}^{het}$ because when $R_{0}^{hom}$ is greater than 1 the EE exit, whereas when $R_{0}^{het}$ is greater than 1 is necessary that $R_{0}^{hom}$ will be greater than 1.

### 2.4. Local Stability of Disease-Free Equilibrium

**Theorem** **5.**
*The DFE $E_{0}=\left(\frac{\Lambda_{h}}{\mu },0,\frac{\Lambda_{w}}{\mu },0,\frac{\Lambda_{m}}{\mu },0,0,0\right)$ is LAS if $R_{0}^{hom}<1$ and $R_{0}^{het}<1$ and is unstable when $R_{0}^{hom}>1$.*


**Proof.** LAS will be demonstrated with the eigenvalues of the Jacobian matrix related to the system (Equation 1) to () evaluated in $E_{0}$, it is:
(29)$$J\left(E_{0}\right)=\left(\begin{array}{cccccccc} -\mu & -B_{h} & 0 & 0 & 0 & 0 & 0 & 0\\ 0 & B_{h}-\left(\alpha +\mu \right) & 0 & 0 & 0 & 0 & 0 & 0\\ 0 & -B_{hw}\frac{1-\theta }{\theta } & -\mu & 0 & 0 & -B_{s}\frac{1-\theta }{\theta \gamma } & 0 & 0\\ 0 & B_{hw}\frac{1-\theta }{\theta } & 0 & -(\alpha +\mu ) & 0 & B_{s}\frac{1-\theta }{\theta \gamma } & 0 & 0\\ 0 & -B_{hm}\gamma & 0 & -B_{s}\frac{\theta \gamma }{1-\theta } & -\mu & 0 & 0 & 0\\ 0 & B_{hm}\gamma & 0 & B_{s}\frac{\theta \gamma }{1-\theta } & 0 & -(\alpha +\mu ) & 0 & 0\\ 0 & \alpha p & 0 & \alpha p & 0 & \alpha p & -(\delta +\mu ) & 0\\ 0 & \alpha (1-p) & 0 & \alpha (1-p) & 0 & \alpha (1-p) & \delta & -(d+\mu ) \end{array}\right)$$The characteristic polynomial is
$$p\left(\lambda \right)={\left(\lambda +\mu \right)}^{3}\left(\lambda +d+\mu \right)\left(\lambda +\delta +\mu \right)\left(\lambda +\alpha +\mu -B_{h}\right)\left[{\left(\alpha +\lambda +\mu \right)}^{2}-B_{s}^{2}\right].$$Eigenvalues are:
(30)$$\begin{array}{cccccc} \lambda_{1} & =-\mu & \lambda_{2} & =-\mu & \lambda_{3} & =-\mu ,\\ \lambda_{4} & =-(d+\mu ), & \lambda_{5} & =-(\delta +\mu ), & \lambda_{6} & =-(\alpha +\mu +B_{s}),\\ \lambda_{7} & =-\left(\alpha +\mu \right)\left[1-R_{0}^{het}\right], & \lambda_{8} & =-\left(\alpha +\mu \right)\left[1-R_{0}^{hom}\right]. \end{array}$$$R_{0}=\max\left\{R_{0}^{hom},R_{0}^{het}\right\}<1$ imply $R_{0}^{hom}<1$ and $R_{0}^{het}<1$; thus, if all eigenvalues are negatives, it follows that $E_{0}$ is LAS. On the other hand, if $R_{0}>1$, then $R_{0}^{hom}>1$ or $R_{0}^{het}>1$, implying that $\lambda_{7}$ or $\lambda_{8}$, respectively, will be positive and in this case $E_{0}$ is unstable. □

### 2.5. Global Sensitivity Analysis

A sensitivity analysis will help us better understand which of the parameters in the model we should focus on estimating most precisely. We answer the following questions: Which parameters contribute most to output variability? Which parameters require additional research or are insignificant? These questions can be answered by performing an analysis with Latin Hypercube Sampling (LHS) and Partial Rank Coefficient (PRCC). We use Matlab to solve the system of ordinary differential equations and to implement most of the SA functions described throughout the manuscript; the functions are available at http://malthus.micro.med.umich.edu/lab/usanalysis.html (accessed on 20 April 2020).

LHS is a statistical sampling method that allows for an efficient analysis of parameter variations across simultaneous uncertainty ranges in each parameter [18]. PRCC shows which parameters have the largest influence on model outcomes [19]. To summarize, we can say that LHS is a sample method, and PRCC conducts the statistical treatment of each sample.

The model contains 12 parameters; however, to perform sensitivity analysis, only parameters related to HIV infection and related to a basic reproduction number are considered. They are: $\gamma ,p,\mu ,\alpha ,B_{h},B_{s},$$B_{hw},B_{hm}$. According to [20], a uniform distribution was chosen over a Gaussian (normal) one because we have no evidence of the ends of the ranges and we carry out multiple runs (NR = 300); parameters, baselines, ranges, and probability density functions (PDF) are listed in Table 2. A Partial Rank Correlation Coefficient was created for each infected population. In addition, scatterplots for each of the aforementioned parameters are presented in Figure 4, Figure 5 and Figure 6.

## 3. Discussion

This analysis focuses on identifying the main parameters that play a dominant role in three different response outputs such as $I_{h}$, untreated infected homosexual men; $I_{w}$, untreated infected women; and $I_{m}$, untreated infected heterosexual men. The more sensitive parameters are: the departure rate of infected individuals, α, and the infection rates in homosexuals and heterosexuals, $B_{h}$ and $B_{s}$, respectively. Scatterplots show the variation in the infected populations size with changes in parameters when examined, thus providing specific qualitative information on the relationship between an infected population and a parameter. Parameters with positive PRCCs will increase $I_{h,w,m}$ when their value is increased, whereas parameters with negative PRCCs will decrease $I_{h,w,m}$ when their value is increased. PRCC values are represented in Figure 4, Figure 5 and Figure 6.

It follows from Figure 4 that untreated infected homosexuals, $I_{h}$, have a negative correlation with the α parameter; in fact, the PRCC = −0.98023, which allows us to conclude that an increase in the α parameter means a decrease in the number of untreated infected homosexual men. In Figure 5, the number of untreated infected women, $I_{w}$, has a positive correlation with rate of infection in heterosexuals $B_{s}$. In fact, PRCC = 0.68725, thus an increase in heterosexual contact implies an increase in women being infected; analogously, this population has a negative correlation with the α parameter. Figure 6 allows us to conclude that the γ parameter does not influence untreated infected heterosexual men, $I_{m}$. In fact, PRCC = −0.0063685, showing that an increase in this parameter has little influence on the number of infected heterosexual men. In addition, infection rates such as $B_{s}$ and $B_{h}$ have similar behavior in infected heterosexual men, $I_{m}$, as for the infected women population, $I_{w}$.

## 4. Conclusions

Models will be a tool for understanding the disease dynamics and for predicting possible trends. Obviously, more accurate predictions require more complex models with more classes and compartments. Although such models are relatively easy to formulate, their mathematical analysis is difficult, and obtaining the necessary social and sexual behavior data is more complicated. Several key features could be included to create more realistic HIV/AIDS models in human populations, such as by looking infectious classes or transmission among injectable drug users through needle sharing.

Sensitivity analysis for the eight parameters related to infection population allow us to conclude that the most influential parameter in the HIV dynamic is the departure rate for infected individuals, α, because it presents the highest PRCC coefficient. This behavior can be explained because the α parameter is present in the basic reproduction number and governs how those infected people are emerging from untreated status to obtain treatment or to develop AIDS.

Bisexual parameters, such as those of the probability of infection via sexual contact between homosexual men and heterosexual men, $\beta_{hm}$, and between homosexual men and women, $\beta_{hw}$, allow us to conclude that higher values of $\beta_{hm}$ and $\beta_{hw}$ imply a high infection rate in untreated infected women and heterosexual men.

Mathematics can provide information for the decision maker about how to promote awareness campaigns aimed at specific populations. This research allowed us to conclude that the best way to reduce contagion and consequently to reach a DFE is thought to be the reduction in homosexual partners rate, as they are the population most affected by the virus and are therefore the most likely to become infected and to spread it. Increasing the departure rate of infected individuals leads to a decrease in untreated infected heterosexual men and untreated infected women. However, it is not the only was to prevent and curb the rate of contagion in San Juan de Pasto. Consequently, it is also necessary to increase anti-retroviral treatment.

With the population parameters of San Juan de Pasto, several numerical simulations were performed by modifying parameters that make the basic reproduction number greater than or less than one. This seems to suggest that when $R_{0}^{het}<1$ and $R_{0}^{hom}>1$, there is a general decline in the rate of HIV infection over the next few years, but the infection persists. As a result, we can conclude that the most important observation from our findings is that, in the population, there is a short-term rise in HIV infection in which there exists a significant increase in new HIV infections, followed by a decline in the generation of new infections.

The dynamic of the HIV/AIDS epidemic, to a large extent, depends on changes in the basic reproduction number among homosexual men, $R_{0}^{hom}$, which was also evidenced by modifying several parameters in the scenarios above. In the background section, it was mentioned that the probability of HIV infection in homosexual men is great than in heterosexual people; thus, the basic reproduction number in heterosexual people, $R_{0}^{het}$, is less influential. In addition, investigations such as [24,25] permit us to conclude that the rate of sexual partners in homosexual men is greater than the rate of sexual partners in heterosexuals; thus, $R_{0}^{hom}>R_{0}^{het}$. This suggests that HIV infection can be controlled or eliminated from the community if control programs are directed towards reducing $R_{0}^{hom}$ to values less than one. The model shows the persistence of the disease when $R_{0}^{hom}>1$.

The dynamics of HIV/AIDS are, in general, too complex to allow for intuitive predictions and require the support of mathematical modeling for qualitatively and quantitatively assessing and understanding the functioning system. Furthermore, one of the most difficult tasks of mathematical modeling is obtaining parameters for a chosen model. Moreover, by using real parameter values to study and analyze the diverse sexual behavior in San Juan de Pasto, the proposed HIV/AIDS model tries to be as approximate as possible to the current situation of this infection. The emphasis was not on the accuracy of the scenarios, but on the actions that can be taken as a result of comprehending the state of the epidemic in the future. For example, scenario 5 shows that when the number of sexual partners is high, the basic reproduction number is greater than 1 and the infection spreads more easily, implying that more and more people are being treated with higher public health costs, and therefore, it is better and more economically efficient to invest in educational campaigns. These actions can involve, among other things, the prevention of new infections, the provision and delivery of anti-retroviral therapy, and educational campaigns such as those that aim to reduce the number of sexual partners or the use of condoms for self-protection.

This application in San Juan de Pasto shows the effects of modifying the parameters related to infected populations. These variations imply huge social and economic expenses which can and should be avoided through government actions such as educational campaigns. In this way, this research aims to be a useful tool in the design of establishing strategies for implementing valid public health policies and introducing efficient public health campaigns.

## Figures and Tables

**Figure 1 viruses-14-02749-f001:**
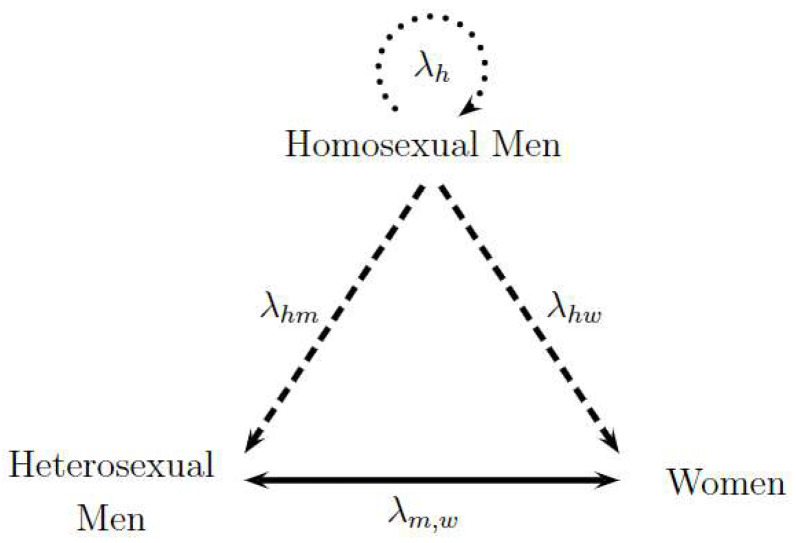
Triangle Transmission in Sexual Preferences: Homosexual Men and Heterosexual Men and Women. Adapted from [15].

**Figure 2 viruses-14-02749-f002:**
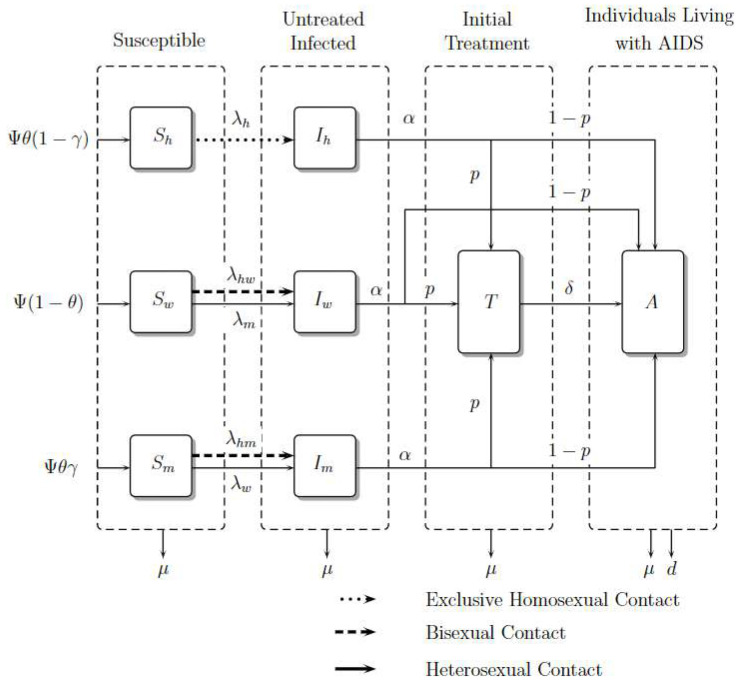
Model Diagram. Adapted from [15].

**Figure 3 viruses-14-02749-f003:**
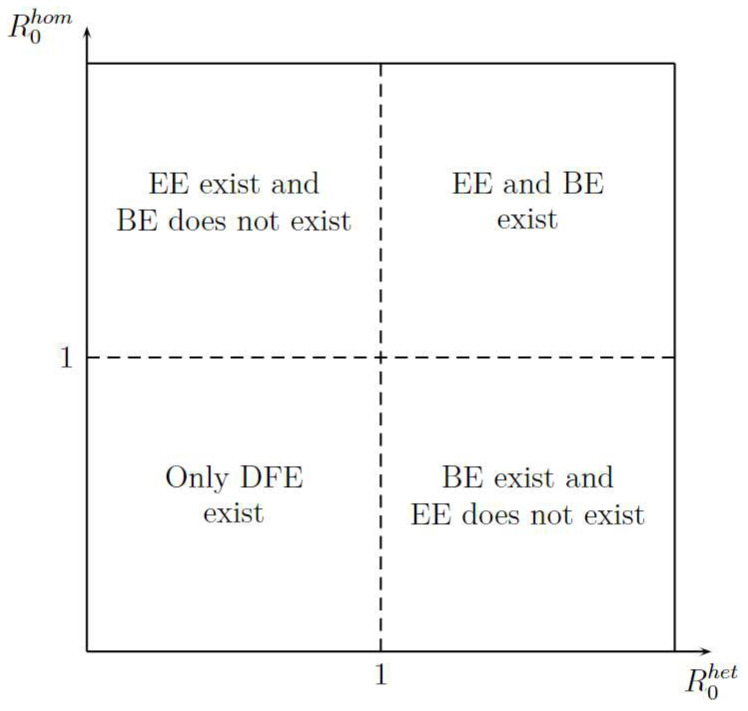
Stationary Points Existence.

**Figure 4 viruses-14-02749-f004:**
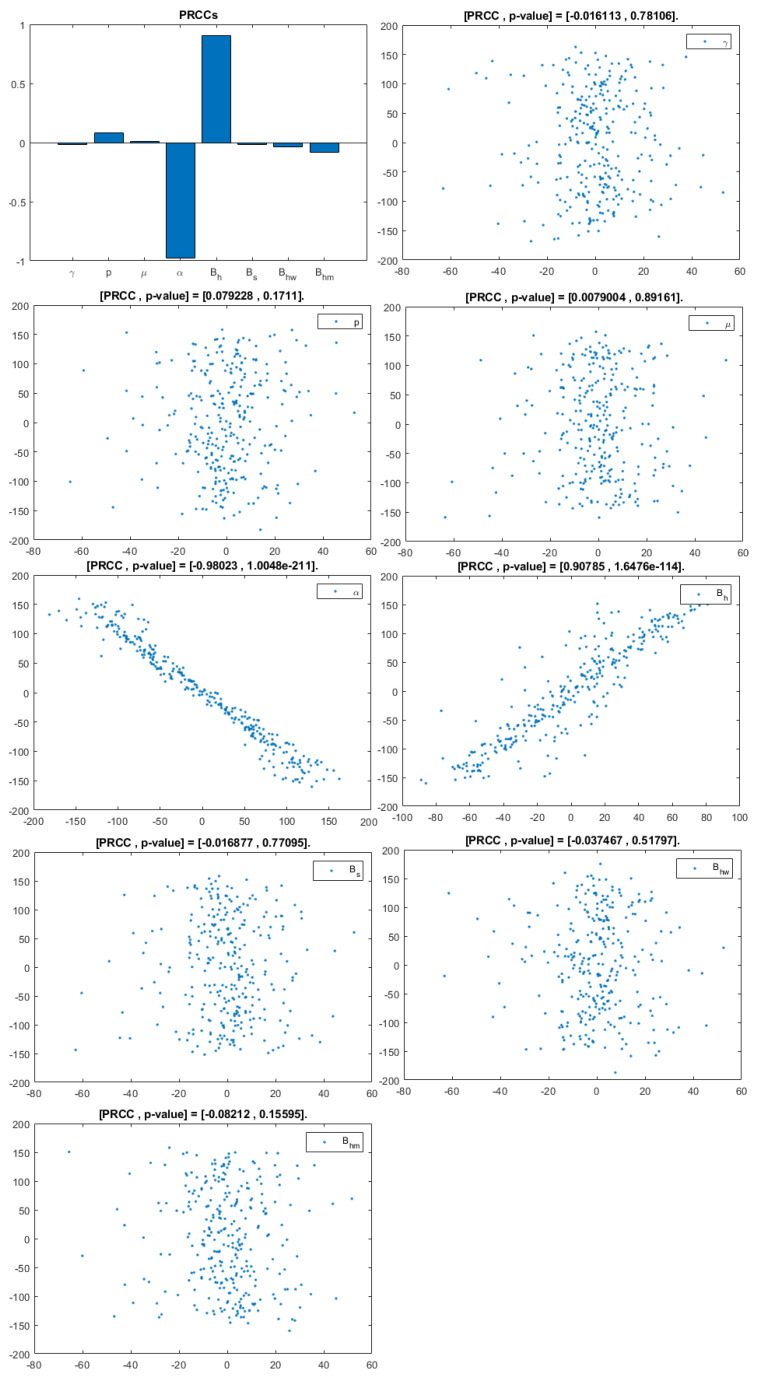
PRCC–Diagram and Scatterplot for each Parameter in Table 2 with respect to Untreated Infected Homosexual Men, $I_{h}$.

**Figure 5 viruses-14-02749-f005:**
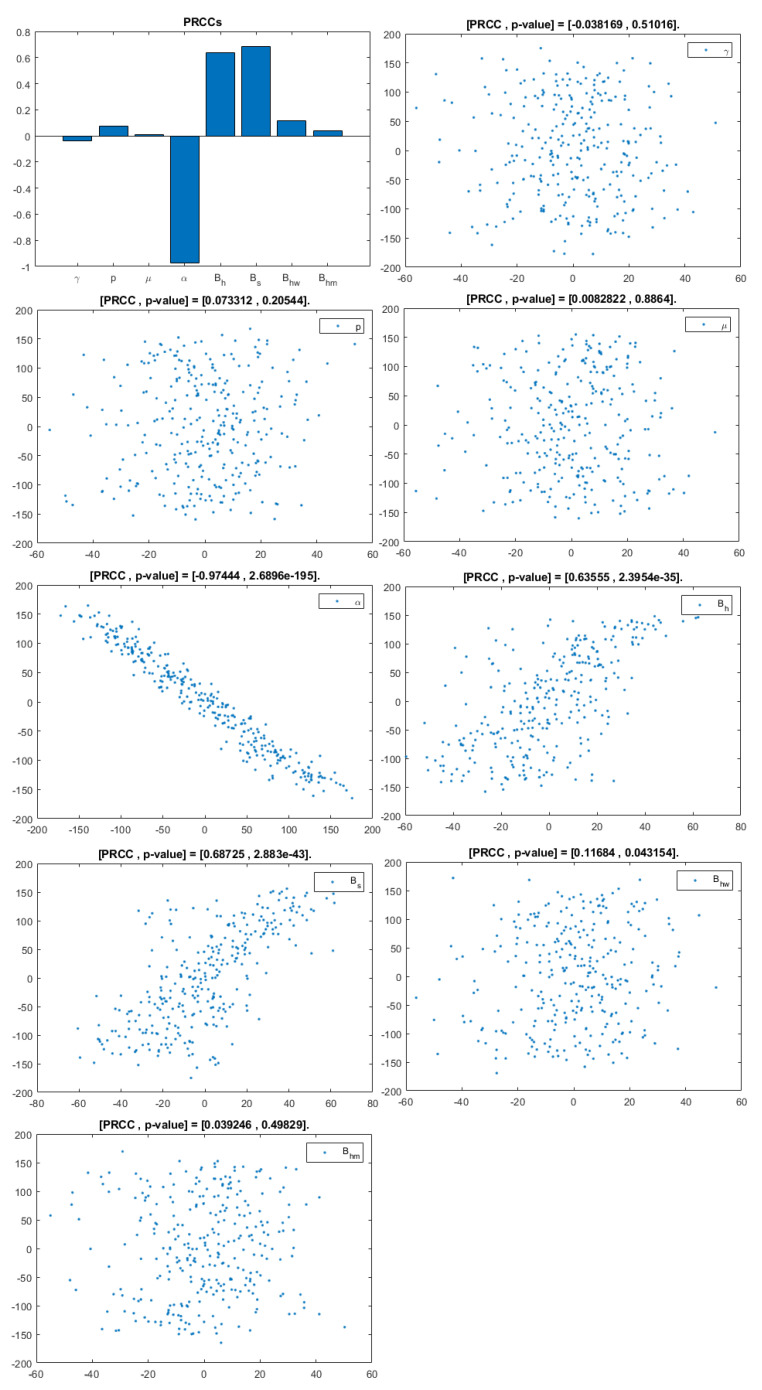
PRCC– Diagram and Scatterplot for each Parameter in Table 2 with respect to Untreated Infected Women, $I_{w}$.

**Figure 6 viruses-14-02749-f006:**
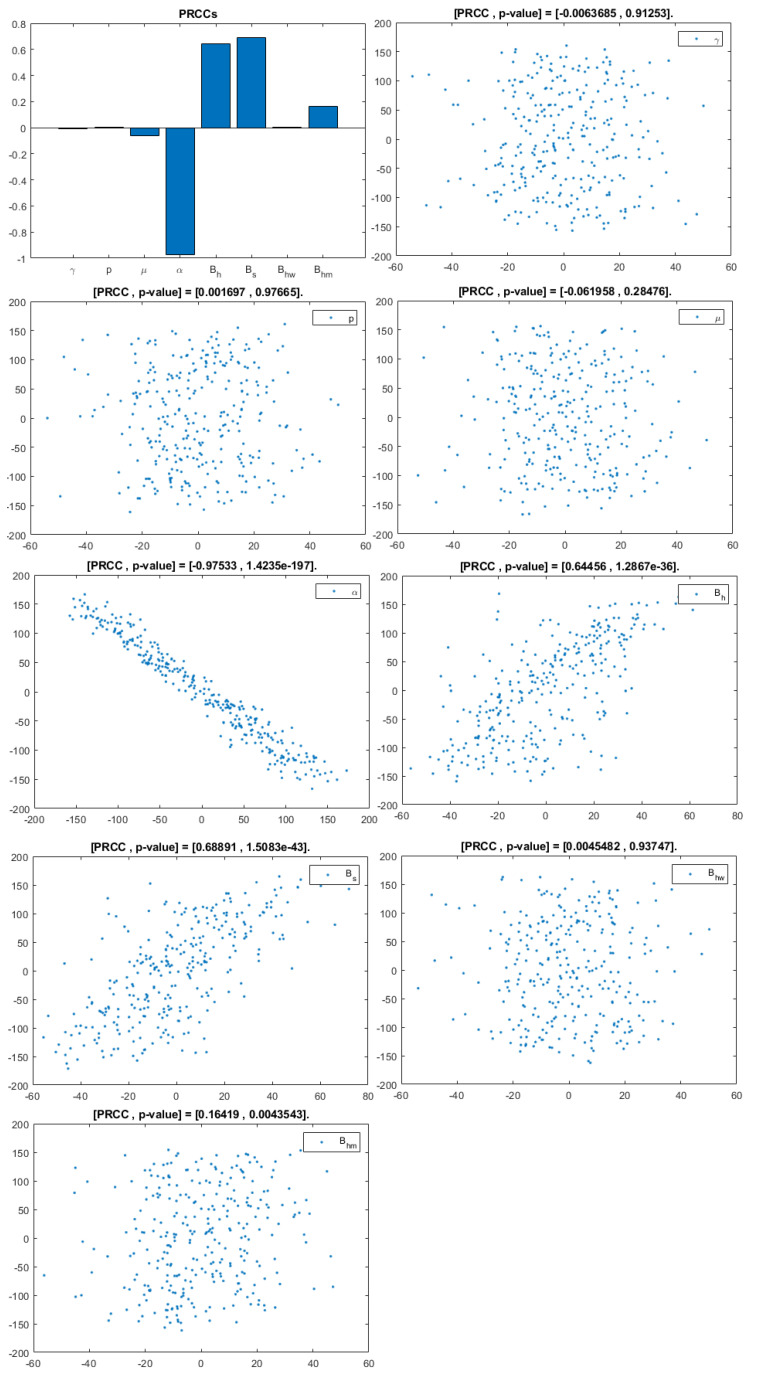
PRCC– Diagram and Scatterplot for each Parameter in Table 2 with respect to Untreated Infected Heterosexual Men, $I_{m}$.

**Table 1 viruses-14-02749-t001:** Description of Parameters. Adapted from [15].

Parameter	Description
Ψ	Constant Recruitment
θ	Male Proportion
γ	Heterosexual Proportion
*p*	Proportion of Initially Treated Individuals
μ	Natural Mortality Rate
*d*	Induced Disease Mortality Rate
δ	AIDS Development Rate in Treated Individuals
α	Departure Rate of infected individuals
$\beta_{s,h,hw,hm}$	Sexual Transmission Probability
$c_{s,h,hw,hm}$	Sexual Partners Rate

**Table 2 viruses-14-02749-t002:** Parameters used in Sensitivity Analysis through Latin Hypercube Sampling and Partial Rank Correlation Coefficients (LHS/PRCC).

Parameter	Baseline	Range	PDF	Source
γ	0.9	[0.3678,1]	Uniform	Assumed
*p*	0.85	[0.1353,1]	Uniform	Assumed
μ	0.0140	[0.01,0.02]	Uniform	[21]
α	0.3333	[0.1353,1]	Uniform	[22]
$B_{h}$	2.64	[0.05,3.95]	Uniform	Assumed
$B_{s}$	0.04	[0.0497,0.5]	Uniform	[23]
$B_{hw}$	0.04	[0.0497,0.5]	Uniform	Assumed
$B_{hm}$	0.3	[0.0497,0.5]	Uniform	Assumed

## Data Availability

Not applicable.

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
