# Peer review of "HIV/AIDS Mathematical Model of Triangle Transmission"

_viruses, 2022, doi:10.3390/v14122749_

Round 1
Reviewer 1 Report
In this paper, mathematical analysis of the HIV/AIDS deterministic model exposed in [Espitia, C. et. al. Mathematical Model of HIV/AIDS Considering Sexual Preferences Under Antiretroviral Therapy, a Case Study in San Juan de Pasto, Colombia, Journal of Computational Biology 29 (2022) 483–493] was studied. The objective is to gain insight into the qualitative dynamics of the model determining the conditions for the persistence or effective control of the disease in the community through the study of basic properties such as positiveness and boundedness, calculus of basic reproduction number, stationary points such as disease free equilibrium (DFE), boundary equilibrium (BE) and endemic equilibrium (EE) are calculated, local stability (LAS) of disease free equilibrium. It research allow to conclude that the best way to reduce contagion and consequently to reach a DFE is thought to be the reduction of homosexual partners rate as they are the most affected population by the virus, and are therefore the most likely to become infected and to spread the infection. Increasing the departure rate of infected individuals, leads to a decrease in untreated infected heterosexual men and untreated infected women.
Before accept,ng this paper should be revised via follows:
*"In this paper, mathematical analysis of the HIV/AIDS deterministic model exposed in [Espitia, C. et. al. Mathematical Model of HIV/AIDS Considering Sexual Preferences Under Antiretroviral Therapy, a Case Study in San Juan de Pasto, Colombia, Journal of Computational Biology 29 (2022) 483–493]"
this sentence may be written as
In this paper, mathematical analysis of the HIV/AIDS deterministic model exposed in [a].
[a] Espitia, C. et. al. Mathematical Model of HIV/AIDS Considering Sexual Preferences Under Antiretroviral Therapy, a Case Study in San Juan de Pasto, Colombia, Journal of Computational Biology 29 (2022) 483–493.
*"2.5. Sensitivity Analysis" should be extended a little more.
*What is the main point of reproducing number?
*How they obtain the equation (A.1)?
*HIV/AIDS was studied in DOI:10.1002/mma.7022; Chaos, Solitons and Fractals, 139(110096), 1-12, 2020. Thus, they need to compare their results.
After these modifications, this paper may be accepted.
Author Response
1.The suggestion for citation in the abstract was accepted and modified it.
2. Section 2.5 was increased.
3. In the triangular model there are two reproduction numbers; for homosexual people and for heterosexual people, it was clearly stated that the main one is the one corresponding to the homosexual population.
4. It is known that the DFE point is obtained by solving the homogeneous system of equations when the infected population is zero. I did not think it necessary to explain this basic concept of epidemiological models. However, I am going to place it.
5. I do not understand what you are referring in the last comment, because there is any mention of the article you mention with doi number.
In the PDF version attached the changes are consigned.

Reviewer 2 Report
The authors explain the difference of this work with Ref[8].
Less contribution from mathematical point of view
The analysis on simulation point of view is nice but should explained in more details.
Simulations regarding disease eliminations must be given.
IN order to fit this submission for publications, the authors need to improve the results both from mathematical and simulations point of view to make it useful for readerships.
The literature must be updated, some literature are here
The dynamics of the HIV/AIDS infection in the framework of piecewise fractional differential equations
Fractional model of HIV transmission with awareness effect
The co-dynamics of hepatitis E and HIV
Author Response
The suggested comments are not written in such a way that the author is asked to modify them, in fact there are some comments that are statements and nothing is asked to be modified. These comments are the first and sixth
There are many comments that ask to make changes in form throughout the writing, forgetting the objective of the article.
For example the second comment asks for less mathematical analysis, but in the abstract the first sentence says that a mathematical analysis will be done. Thus I do not think it is appropriate to follow this suggestion since it implies changing the objective.
The third, fourth and fifth comments are very general, since in the paper it was never mentioned that numerical simulations will be done, it was said that only parameter analysis is done.
The sixth, seventh and eighth comments are totally wrong, since the model presented is not of fractional calculus or co-dynamics with any other disease. In this regard, I believe that these comments are for another research.